# Investigation on the Molecular and Physicochemical Changes of Protein and Starch of Wheat Flour during Heating

**DOI:** 10.3390/foods10061419

**Published:** 2021-06-18

**Authors:** Tao Yang, Pei Wang, Qin Zhou, Xiao Wang, Jian Cai, Mei Huang, Dong Jiang

**Affiliations:** 1College of Agriculture, Nanjing Agricultural University, No. 1 Weigang Road, Nanjing 210095, China; 2017101046@njau.edu.cn (T.Y.); xiaowang@njau.edu.cn (X.W.); caijian@njau.edu.cn (J.C.); huangmei@njau.edu.cn (M.H.); jiangd@njau.edu.cn (D.J.); 2College of Food Science and Technology, Nanjing Agricultural University, No. 1 Weigang Road, Nanjing 210095, China

**Keywords:** heating, wheat flour, protein, starch, interactions, viscosity, texture

## Abstract

The behaviors of starch and protein in wheat flour during heating were investigated, and the molecular changes of starch and protein and their effects on the textural characteristics were assessed. The results showed that with the increased temperature, soluble protein aggregated to insoluble high-molecular-weight protein polymers when the heating temperature exceeded 70 °C, and the aggregation of protein was mainly caused by covalent bonds of disulfide (SS) bonds. Hydrophobic interaction was the main noncovalent bond that participated in the formation of protein aggregates. The major change in the secondary structure during heating was a pronounced transition towards β-sheet-like structures. Considerable disruption of ordered structures of starch occurred at 70 °C, and starch was fully gelatinized at 80 °C. Typical starch pasting profiles of cooked flour were observed when the temperature was below 70 °C, and heat treatment decreased the pasting viscosity of the cooked flour from control to 80 °C, whereas the viscosity of the wheat flour increased in heating treatment at 90, 95 and 100 °C. The intense protein-starch interaction during heating affected the textural characteristic of flour gelation, which showed higher strength at 90, 95 and 100 °C. This study may provide a basis for improving wheat flour processing conditions and could lead to the production of new wheat products.

## 1. Introduction

Wheat (*Triticum aestivum* L.) is one of the major constituents of the human diet because of its various end uses; it can be used to make a wide range of foods such as biscuits, bread and noodles [1]. Starch is the predominant component in wheat grain, which is composed of two types of glucose polymers, amylose and amylopectin, and is present in different granule sizes. Protein is the second largest component in wheat grain, which is classified into two fractions according to its solubility in alcohol-water solutions: insoluble glutenins (about 40–50%) and soluble gliadins (about 50–60%). Glutenins are interchain disulfide (SS)-linked polymers with a wide molecular weight (Mw) distribution from 10^5^ to 10^7^ Da and impart the elasticity of dough. Two distinct groups of subunits can be distinguished: HMW-GS and LMW-GS. The majority of gliadins are monomeric proteins with Mw ranging from 3 to 8 × 10^4^ Da and confer viscosity to the dough. Gliadins can be further classified into α-, ω- and γ-type subgroups. Heating is one of the most important processes in wheat flour processing, and the heat-induced physicochemical evolution of flour components is a complex process. It can improve food texture and hold moisture or functional components, which in turn affects the flavor, color and texture of the final product [2]. Knowledge of the structure and physicochemistry changes of protein and starch during heat processing is important to understanding the mechanism of protein–starch interactions during the processing of flour as well as for the manufacture of wheat-based foods.

Protein and starch have complex hierarchical structures which are studied with multiple techniques. For example, the differences in the structure of gluten in different forms can be characterized by secondary, tertiary and spatial structure by Fourier transform infrared spectroscopy (FTIR), intrinsic fluorescence spectra and confocal laser scanning microscopy (CLSM). Meanwhile, FTIR, X-ray diffraction (XRD) and rapid viscosity analysis (RVA) can be used to determine the structure of starch from a nano-to-micrometers perspective [3,4]. During heat processing, starch granules are gelatinized and proteins are cross-linked, causing disruption of the multi-scale structure, with the extent of disruption depending on temperature and a series of heat processing conditions. Interactions between starch, proteins and other components during heat processing have been studied extensively in model systems [5,6]. Wang et al. [7] extracted gliadin from wheat flour and found that gliadin polymers were formed at 90 °C and resulted in the loss of ethanol extractability. Wang et al. [8] studied changes in chemical interactions during heat-induced wheat gluten gel formation, and found that crosslinking reactions among the wheat gluten molecules occurred when the temperature was higher than 60 °C. Liu et al. [9] explored the disassembly mechanisms of starch granules during thermal food processing using purified wheat starch as a model material, and found the gelatinization and short-range molecular order of wheat starch were affected greatly by the heating temperature and water content.

The presence of multiple components in wheat-based food systems may affect the behavior of starch or protein during processing through the potential for complex formations [10,11]. The presence of multiple components in flour is assumed to affect the functionality of protein and starch in food systems. For example, the presence of proteins and other components in cereal-based food systems may affect the dynamic behaviors of starch or protein during heat processing through the potential for complex formation with them [12]. However, there is little actual information on this and most research has been conducted on unitary studies of separated protein or starch. Therefore, study of the behavior of protein or starch which is isolated from the complete flour system cannot completely reflect the change and interaction of components in wheat-based foods during heat processing. For example, how protein and starch influence the behaviors of each other during the cooking of doughs and batters still remains unclear. Studying the mechanism of protein and starch behavior during wheat flour heating is an important step to predict and control the processing and functional properties of more complex cereal food systems. Therefore, in this study, a heating model system of wheat flour was used to study the molecular and physicochemical changes of starch and protein during heat processing.

## 2. Materials and Methods

### 2.1. Materials

Winter wheat (*Triticum aestivum* L.) cultivar Yangmai 22 was planted at the Center of Rice and Wheat Science and Technology demonstration in Jintan, Jiangsu province, China. The plant density was 1.6 × 106 ha^−1^. Fertilizers of N, P and K were applied as 180 kg N ha^−1^, 120 kg P_2_O_5_ ha^−1^ and 120 kg K_2_O ha^−1^, respectively. The ratio of basal to topdressing of N was 6:4, and topdressing fertilizer was applied at jointing stage.

At maturity, grains of Yangmai 22 were harvested and cleaned. After tempering to 14% moisture for 18 h, wheat grains were milled using a flour miller (ZS70-II, grain and oil foodstuff machine factory, Zhuozhou, China) with a 100μm mesh sieve. Flour was stored in refrigerator (4 °C) for further analysis.

### 2.2. Preparation of Flour Samples Heated with Different Temperature

In order to eliminate the influence of water, wheat flour (3.5 g) was weighed accurately into an aluminum case, and distilled water (25 g) was added to form wheat flour dispersions. Subsequently, the dispersions were homogenized for 30 s, then stirred continuously with a shear speed of 160 rpm/s. Meanwhile, the dispersions were incubated at 60, 70, 80, 90, 95 and 100 °C for 30 min, respectively, and the flour dispersions at room temperature (about 30 °C) were set as control. The dispersions treated with different temperatures were sampled and stored in liquid nitrogen. After lyophilizing, the samples were milled for further analysis.

### 2.3. Methods

#### 2.3.1. Size Exclusion-Fast Protein Liquid Chromatography (SE-HPLC)

Freeze-dried samples (10 mg) were extracted with 1.0 mL of 0.05 M sodium phosphate buffer (pH 6.8, containing 2% SDS (*w*/*v*)) for 1 h at room temperature. After centrifugation (5000× *g*, 4 °C, 5 min), 20 μL of the supernatant was filtered by 0.45 μm membrane, and then loaded on a Shodex Protein KW-804 column (Showa, Kyoto, Japan), which is suitable for the analysis of proteins with molecular weight ranging from several thousand to several million. The elution was achieved with a 0.2% sodium phosphate buffer (0.05 M, pH 6.8) at 30 °C with flow rate of 0.7 mL/min. Eluted protein was detected at 214 nm. The solution of reduced sample contained 1% DTT. The analyzed parameters were calculated as follows:SDS-P = Ap/Ar × 100(1)
SDS-M = Am/Ar × 100(2)
SDS-I = 100 − (SDS-P + SDS-M)(3)

In the above formulas, SDS-P, SDS-M and SDS-I refer to SDS soluble polymers proteins, SDS soluble monomers proteins and SDS-insoluble proteins. Ap and Am are the peak areas of polymers and monomers extracted by 0.05 M sodium phosphate buffer, respectively. Ar was the peak area of reduced proteins which was extracted with reduced agent (1% DTT) [13].

#### 2.3.2. Sodium Dodecyl Sulfate Polyacrylamide Gel Electrophoresis (SDS-PAGE)

SDS-PAGE analysis of proteins was performed in a vertical electrophoresis cell using 10% separating gel and 6% stacking gel. Each sample (50 mg) was dissolved in 1.5 mL of extraction buffer (Tris-HCl, 0.125 M, pH 6, containing 2% SDS (*w*/*v*), 10% glycerol (*v*/*v*) and 0.01% bromophenol blue (*w*/*v*)) and was left for 3 h at room temperature. For reduced proteins, the extraction buffer contained 5% (*v*/*v*) β-mercaptoethanol (β-ME). After centrifugation (10,000× *g*, 4 °C, 20 min), supernatant was heated for 5 min in boiling water and loaded after cooling.

#### 2.3.3. Determination of Free Sulfhydryl (SH) Group

Free sulfhydryl content was determined according to the method of Lambrecht et al. [14] with minor modification. A freeze-dried sample was mixed with 4 mL SDS-TGE buffer (pH 8.0, 2.5% SDS, 86 mM Tris-HCl, 92 mM glycine and 4.1 mM EDTA), and was homogenized for 30 s, then left for 30 min with homogenizing intermittently every 10 min 3 times. 40 μL Ellman’ reagent (DTNB in SDS-TGE buffer, 4 mg/mL) was added and the tubes were wrapped with aluminum foil, then left at room temperature for 30 min. After centrifugation (20,000× *g*, 4 °C, 15 min), the absorbance of supernatant was detected at 412 nm.

#### 2.3.4. Determination of Non-Covalent Bonds of Protein

Chemical reactions were determined according to the method described by Wang et al. [15]. Selective buffers prepared in phosphate buffer (0.05 M, pH 7.0) were used to solubilize proteins as follows: 0.05 M NaCl (S1), 0.6 M NaCl (S2), 0.6 M NaCl + 1.5 M urea (S3) and 0.6 M NaCl + 8 M urea (S4). Flour samples (100 mg) was extracted with the above four agents (10 mL) at room temperature for 1 h. After being centrifuged (10,000× *g*, 4 °C, 20 min), the soluble protein content in the supernatant was determined by micro-Kjeldahl method with conversion factor of 5.7. The difference of soluble protein between S1 and S2, S2 and S3, S3 and S4 reflected ionic bonds, hydrogen bonds and hydrophobic interactions, respectively.

#### 2.3.5. Fourier Transform Infrared Spectroscopy (FTIR)

FTIR spectroscopy was used to measure infrared spectra in the region from 400–4000 cm^−1^ using a Thermo Nicolet Nexus FTIR (Thermo Scientific, Waltham, MA, USA) with a single-reflection diamond attenuated total reflection (ATR) crystal and a mercury-cadmium-telluride (MCT) detector. Background spectrum of ATR was recorded at 64 scans and 4 cm^−1^ resolution, and FTIR spectra of each sample was recorded under the same condition against the background. The amide I band (1600–1700 cm^−1^) in spectrum was deconvoluted after baseline-correction and Gaussian smooth to quantify the secondary structure of protein using Omnic software (version 6.1a, Thermo Nicolet Corp., Madison, WI, USA) and Peakfit software (version 4.12, SPSS Inc., Chicago, IL, USA). The spectrum ranged from 980–1060 cm^−1^ and was deconvoluted using Omnic software (version 6.1a, Thermo Nicolet Corp., Madison, WI, USA). The bands at 1047 and 1022 cm^−1^ were detected; the ratio of the integrated area of absorption bands at 1047/1022 cm^−1^ is generally used to quantify the internal changes of the starch molecule in the degree of short-range order [16].

#### 2.3.6. Intrinsic Fluorescence Spectra

The intrinsic fluorescence spectra were determined by LS-55 luminescence spectrometer (Perkin Elmer Life Sciences, Shelton, 181 CT, USA) equipped with the Perkin Elmer FL Winlab software. The sample (25 mg) was dissolved in 5 mL 50 mM acetic acid solution at room temperature for 1 h and subsequently centrifuged (10,000× *g*, 15 min). The supernatant was diluted to 1 mg/mL with extraction solution. The excitation wavelength was set as 280 nm and the emission spectra were recorded from 300–420 nm with a 1 nm slit. The intrinsic fluorescence spectroscopy was used to depict the alterations in protein tertiary structure due to the sensitivity of the protein amino acid residues to the polarity of the microenvironment.

#### 2.3.7. Pasting Properties

Pasting properties were analyzed with Rapid Viscosity Analyzer 130 (RVA-3D super type, Newport Scientific, Stockholm, Sweden) following the method described by Chang et al. [17] with some modifications. A freeze-dried sample (1.5 g, on 14% dry basis) was suspended in deionized water (25 mL). The procedure of RVA included a heating step, a holding step and a cooling step. During heating, temperature increased from 30 °C to 40 °C for 1 min, and then linearly increased to 95 °C at a rate of 6 °C/min, then held at 95 °C for 5 min. During cooling, the temperature linearly decreased to 50 °C at a rate of 7.5 °C/min, and finally held at 50 °C for 3 min. The parameters of peak viscosity, trough viscosity, breakdown, final viscosity, pasting temperature and peak time were recorded.

#### 2.3.8. Relative Crystallinity of Starch Determined by X-ray Diffraction (XRD)

Samples were sieved by 320 μm sieves after grinding by mortar prior to XRD. The X-ray diffractometer (TD-3500, Tongda, China) was operated at 40 KV and 40 mA with Cu Kα radiation (λ = 0.154 nm). The sample was packed tightly in a glass cell and scanned over the range of 5–40° 2θ angles at 2°/min at room temperature. The relative crystallinity of the starch was calculated using the following formation:Relative crystallinity(%) = (100 × Ac)/(Ac + Aa)(4)
where Ac is the crystalline area on the X-ray diffractogram, and Aa is the amorphous area.

#### 2.3.9. Texture Characteristics

Texture profile analysis was performed using a texture analyzer (TA. XT2i, Stable Micro Systems, Surrey, UK) using a P/0.5 probe at room temperature. The selected settings were 1.5 mm/s of pre-test speed, 1.0 mm/s of test speed and 1.0 mm/s of post-test speed with trigger force of 3 g. The force of probe puncture to 4 mm was expressed as bloom strength. The force of first peak is expressed as the rupture strength, which can reflect the parameters of the process or how much force is required to bite when chewing. Adhesiveness was determined by the area of the domain formed by force and time when the probe returned. 

#### 2.3.10. Confocal Laser Scanning Microscopy (CLSM)

A lyophilized sample was mixed with water to form dough organization, and was cut into 20 μm slices by freezing microtome (Leica CM3050S, Leica Bioystems Nussloch Gmbh, Germany). Slices were stained with Rhodamine B agent (0.001%, *w*/*v*) to observe the protein network in the dough by CLSM (ZEISS LSM800, Oberkochen, Germany). Parameters of protein network such as protein area (μm^2^), junction density (×10^−3^), total protein length (μm) and lacunarity (×10^−2^) were analyzed using AngioTool version 0.5 [18] according to Bernklau et al. [19].

### 2.4. Statistics Analysis

All data were expressed as mean ± standard deviation (SD) of three replicates. Data was analyzed using one-way analysis of variance (ANOVA), and Duncan’s multiple range test was used to compare the means with a significance using an SPSS package (version 10.0 for Windows, SPSS Inc. Chicago, IL, USA). The probability value of *p* < 0.05 was considered significant.

## 3. Results

### 3.1. SDS-PAGE Analysis

Non-reduced and reduced SDS-PAGE was conducted to verify the formation of disulfide (SS) bonds during heat treatment (Figure 1). The electrophoresis patterns can be divided into polymerized protein (130–200 KDa), high-molecular-weight glutenin subunit (HMW-GS) (75–120 KDa), ω-gliadin (50–75 KDa), low-molecular-weight glutenin subunit (LMW-GS) (36–44 KDa) and α/β/γ-gliadin (28–43 KDa). From the non-reduced SDS-PAGE, the band density of soluble polymerized protein, HMW-GS and LMW-GS decreased continuously with the increase of temperature. The band of polymerized protein and HMW-GS was clearly seen under treatment of 60 and 70 °C, while it gradually decreased when the temperature was higher than 80 °C, and the bands of SDS-soluble polymerized protein disappeared at 100 °C. When SDS-soluble protein was reduced by β-ME, the bands of SDS-soluble polymerized protein disappeared. At the same time, the bands density of HMW-GS and LMW-GS increased with the increase of temperature.

### 3.2. Distribution of Protein Molecular Weight (Mw)

In the non-reduced SE-HPLC chromatograms of proteins (Figure 2A-1), the area of the peaks of SDS-soluble polymers and monomers of protein decreased with the increase of temperature. After cleavage of SS bonds with the addition of DTT, polymer proteins were reduced to monomer proteins and a major peak mainly appeared at ~13 min (Figure 2A-2). SDS-M, SDS-P and SDS-I levels are shown in Figure 2A-3. The levels of SDS-M (soluble monomer protein) and SDS-P (soluble polymers protein) all decreased markedly with the increase of temperature. On the contrary, the SDS-I (insoluble protein) level increased significantly with the increase of temperature, and increased largely when the temperature exceeded 70 °C.

### 3.3. Analysis of Intermolecular Interactions

Intermolecular interactions involved in protein polymerization during heating are mainly free SH, disulfide and noncovalent bonds. As show in Figure 2B, the free SH content continuously decreased from 5.36 μmol/g to 2.83 μmol/g as the temperature increased from room temperature to 100 °C, with 19.86% reduction from room temperature to 60 °C.

The protein solubility of all samples increased as the concentration of NaCl (S2) increased (Figure 2C), and further increased with the addition of urea (S3), indicating that ionic bonds and hydrogen bonds participated in the formation of protein spatial structure. In addition, the protein solubility of all samples increased greatly with the addition of 8 M urea (S4), indicating that hydrophobic bonds played an important role in the formation of protein aggregates. The protein solubility of proteins in S1–S4 decreased with the increase of temperature, which indicated that ionic bonds, hydrogen bonds and hydrophobic bonds were destroyed or weakened due to the formation of protein aggregates at higher temperatures.

### 3.4. Intrinsic Fluorescence Spectra Analysis

The intrinsic fluorescence spectroscopy was performed to obtain information concerning protein tertiary structure changes (Figure 2D). The intrinsic fluorescence intensity reduced largely when temperature exceeded 70 °C and reached its minimum at 100 °C. It was observed that the maximum wavelength (λmax) of all samples ranged from 339–350 nm. Red-shifts in λmax of 2 nm were observed from CK to 70 °C, and the differences between CK, 60 and 70 °C were not significant, while the λmax significantly blue-shifted when the temperature exceeded 70 °C and reached its minimum at 100 °C.

### 3.5. Secondary Structure Analysis

The FTIR spectra of samples are shown in Figure 3A-1. All samples had similar absorption peaks in the region of 500–4000 cm^−1^. A broad band located at approximately 3400 cm^−1^ and a band at around 2950 cm^−1^, typically presented in polysaccharides and assigned to O–H stretching of the inter hydrogen-bond and C–H stretching of methylene, were observed, respectively. The intensity of these bands increased with the increase of temperature, which indicated that heat improved the hydration. An additional peak at about 2850 cm^−1^ occurred due to CH and CH_2_ stretching in cellulose. During heat processing, the band intensity at around 2850 cm^−1^ increased with the increase of temperature, indicating that the peak of polymer hydroxyl stretching vibration was enhanced, and the greater intermolecular interactions were formed during heating. The peak at 1746 cm^−1^ had been identified as the ester carbonyl group; it is noteworthy that this peak was more pronounced at a higher temperature, probably due to the formation or reinforcement of the starch-lipid complex.

The content of the α-helix and intermolecular β-sheets increased with the increase of temperature, while the contents of the β-sheets decreased. Besides, the content the of anti-parallel β-sheets, β-turn, fluctuated during the heating of the wheat flour (Figure 3A-2). Among them, the contents of the α-helix and intermolecular β-sheets changed greatly with the increase of temperature. When heated from 60 to 100 °C, α-helix increased 13.2%. Control had higher intermolecular β-sheets contents, and intermolecular β-sheets contents increased dramatically with an increase of 71.58% as the temperature increased from 60 °C to 100 °C.

### 3.6. Starch Crystallization

The XRD patterns and crystallinity of starch were influenced by temperature largely (Figure 3C). The diffractograms of the control were typical A-type crystal structures with strong reflection peaks at 15°, 17°, 18° and 23°. The number of peaks decreased with the increase of temperature, and there was only one peak remaining when the treatment temperature exceeded 70 °C. The relative crystallinity decreased from 17.10% to 1.40% with increasing the temperature, and control had the highest relative crystallinity.

The deconvoluted FTIR spectra ranged from 1060 to 980 cm^−1^ as shown in Figure 3B-1. The spectrum of 1047 and 1022 cm^−1^ corresponded to the characteristic of crystallization and amorphous of starch granule, respectively. The IR ratio of 1047/1022 cm^−1^ decreased with increasing temperature, especially when the temperature increased from room temperature to 60 °C (Figure 3B-2).

### 3.7. Starch Pasting Properties

Starch pasting properties were determined after the heating treatment of wheat flour. Pasting properties graphs of samples during heating are shown in Figure 3D. The control showed the highest peak viscosity, and the peak viscosity reached its minimum at 80 °C. Besides, peak viscosity at 90, 95 and 100 °C increased compared with viscosity at 80 °C, while the peak viscosity at 100 °C was still far less than control. The pasting temperature increased with the increase of the heat treatment temperature from room temperature to 80 °C, while pasting temperature at 90, 95 and 100 °C was lower than pasting temperature at 80 °C.

### 3.8. Texture Profile Analysis

The texture profiles and parameters of the different treatments are shown in Figure 3E. When the probe was puncturing, the force increased when treatment temperature increased. The force of treatments at 90, 95 and 100 °C was significantly higher than those in treatment at 60, 70 and 80 °C. Gelation strength and rupture strength remarkably increased with incensement of 82.8% and 104.3% when the temperature increased from 80 °C to 90 °C. Adhesiveness exhibited a significant reduction when the temperature was below 90 °C, while the differences between 60, 70 and 80 °C were not significant.

### 3.9. Morphological Properties

The protein network was dyed red by non-covalent binding with Rhodamine B (Figure 4). The protein network in control, in which round or oval starch granules were embedded, was more continuous and uniform. With the increase of temperature, the structure of the protein network rearranged and aggregated. In treatments of 95 and 100 °C, the protein aggregated into more amorphous which were uneven and more compact. To further explore the protein aggregation during heating, the protein networks were quantified by the software of Angiotool (Table 1). The junction density, total protein length, branching rate and lacunarity showed an increasing trend with the increase of temperature. Under treatment of 95 °C, the lacunarity increased by 57.5% as compared to treatment of 90 °C. The protein area decreased dramatically with the increase of temperature, especially when increased from 95 to 100 °C.

## 4. Discussion

In this study, the molecular and physicochemical changes of protein and starch in wheat flour suspension during heating were monitored by a combination of analytical methods. The advantage of using wheat flour as a model system to study dynamic changes of protein or starch compared to pure starch or protein during the heating process or simulated food processing is that it can reveal the influence of other ingredients (fat, vitamins, minerals, etc.) in the native flour. Therefore, this study is related to foods made from dough (e.g., baked products, noodles, pasta) and batters (e.g., pancakes, coatings for fried foods). The results showed that the band density of SDS-soluble polymers diminished gradually with the increase of temperature in non-reduced SDS-PAGE profiles, especially when the temperature exceeded 80 °C, which indicated that protein aggregated at higher temperatures. The band density of HMW-GS and LMW-GS in reduced SDS-PAGE increased at higher temperatures, indicating that more HMW-GS and LMW-GS participated in protein aggregation. The difference between non-reduced and reduced SDS-PAGE showed that polymerized protein could be reduced by β-Me, which showed that the SS bond was the main intermolecular covalent bond in aggregated protein. The decrease of free SH levels coincided with the increase of SS bonds, which further demonstrated that new SS bonds should be formed by the oxidation of SH [20]. In addition, the absorption peaks of at 2850 cm^−1^ indicated the interaction of the mercaptan group and the intermolecular hydroxyl group, and the band intensity increased with the increase of temperature, which indicated that it was easier to form SS bonds from mercaptan groups of sulfur-containing amino acids under a higher temperature, which further demonstrated that higher temperatures facilitated the formation of SS bonds by thiol end functional groups in sulfur-containing amino acids [21]. SEC-HPLC is an important method for protein size fractionation and extractability. In this study, monomeric protein vanished with the DTT addition while the peak of polymeric protein was retained (Figure 2A-2), indicating that SDS-soluble monomeric proteins participated in the formation of SDS-soluble protein polymers through the intermolecular SS bond [7]. Meanwhile, SDS-insoluble protein levels increased with the increase of temperature and increased greatly when the temperature exceeded 70 °C, indicating the formation of protein aggregates at higher temperatures, and this was in line with the results of SDS-PAGE and free SH content. Apart from SS bonds, interactions of non-covalent bonds such as hydrogen bonds, ionic bonds and hydrophobic interactions also played essential roles in maintaining the three-dimensional spatial structure of the protein network. According to the results of protein solubility in different agents (Figure 2C), the ionic bond and hydrogen bond participated in the formation of the protein chain, but they were easily destroyed during the heating of the wheat flour. The destroyed ionic and hydrogen bonds caused a lot of exposure of hydrophobic groups and formed a hydrophobic interaction, which contributed the most in maintaining protein spatial structure.

In addition, heat treatment induced the fluctuation of a protein secondary structure, α-helix and intermolecular β-sheets content increased, while β-sheets decreased with the increasing temperature. An especially significant decrease was observed at 70 °C, which was consistent with the results of Francesco et al. [22]. The increase in α-helix meant the formation of ordered and compact protein spatial structure [23] by enhancing protein folding [24]. The increase in intermolecular β-sheets further demonstrated that protein polymerized during heating of wheat flour. In order to obtain further information concerning protein structural changes, an intrinsic fluorescence spectroscopy was used to depict the alterations in protein tertiary structure [25]. The intrinsic fluorescence intensity reduced largely when the temperature exceeded 70 °C and reached its minimum at 100 °C. This was mainly attributed to a shift in the Trp microenvironment toward a more hydrophilic microenvironment, indicating that the extent of protein polymerization was enhanced. Besides, red-shifts in λmax of 2 nm were observed from CK to 70 °C, indicating that the Trp residues were exposed to a solvent, and the peptide strands became extended, which is an indicator of protein unfolding [26]. The λmax significantly blue-shifted when the temperature exceeded 70 °C and reached its minimum at 100 °C, further suggesting that Trp was exposed in the hydrophilic microenvironment.

During heating, starch granules also changed greatly: the long-range order and short-range order of the starch structure were determined. XRD was used to analyze the regular arrangement of the long-range ordered double helix structure of starch [27]. The treatment of control displayed a typical A-type pattern with four major signal peaks and showed the highest crystallinity. When the temperature increased to 60 °C, the intensity of the four reflection peaks decreased slightly and relative crystallinity decreased as compared with control indicating the slight degradation of the crystalline structure [28]. When the temperature exceeded 60 °C, the reflection peaks of the samples changed from A-type patterns into V-type, the typical A-type pattern disappeared, and there was a low and wide dispersion peak, which can be interpreted as the destruction of starch crystallization due to the movement of amylopectin double helices [29]. From the results of the XRD, it could be seen that under high temperature treatment the starch was completely gelatinized, and the long-range ordered structure of starch had been completely destroyed. Therefore, the short-range order of the starch structure was determined by infrared spectrum. The infrared spectrum is very sensitive to the short-range ordered structure of starch granules and can distinguish the short-range ordered structure [30]. The absorbance of 1047 and 1022 cm^−1^ detected by FTIR corresponded to the crystalline and amorphous structure of starch, respectively [31]. In our study, the IR ratio of 1047/1022 cm^−1^ decreased with the increase of temperature indicating that the starch granule structure was destroyed, and more amorphous structure was formed with increasing the temperature.

Different heat treatments had different effects on the peak viscosity. The viscosity decreased as the temperature of the heating treatment increased from control to 80 °C. Amylose acted as a diluent or an inhibitor of swelling and even formed insoluble complexes with wheat lipid which restricted swelling during starch gelatinization [5]. In addition, heat-induced protein aggregates acted as an inhibitor of starch swelling [6], and the results were in line with previous research [12,32]. Meanwhile, the denser and more compact morphology of the gluten network at higher temperatures observed by CLSM (Figure 4) demonstrated the role of protein aggregates as barriers, and the faded band of soluble polymerized protein in non-reduced SDS-PAGE (Figure 1) and higher levels of SDS-I and blue-shifted emission wavelength at higher temperatures also confirmed that. Therefore, interactions between denatured protein and starch delayed the starch swelling and caused the decrease of viscosity. Interestingly, although the overall crystallinity of starch was completely disrupted at higher temperatures, the existence of a starch short-range ordered structure made starch expand, which caused the heated sample to develop the typical viscosity pattern. While viscosity increased when the temperature of heat treatment exceeded 80 °C, this was related to the presence of cold water-swelling starch. Cold water-swelling starch, sometimes called pregelatinized starch, had complete degradation of the starch structure, and similar results were also found in rice [33]. These results indicated that starch, especially amylose, and protein in wheat flour prevented the starch granules from swelling which led to a decrease in peak viscosity at higher temperatures.

Flour gelation can be obtained by cooling flour pasta after gelatinization. During starch retrogradation, the rheological properties and crystallinity of the gelation change significantly, and this is the main factor that affects food texture. Meanwhile, starch retrogradation is based on the degree of starch gelatinization. Accordingly, flour gelation is the result of starch gelatinization and protein aggregation, and it is the macroscopic manifestation of micro level changes of protein and starch during the heating of wheat flour. In our study, gelation strength and rupture strength increased with the increase of treatment temperature, especially when the temperature exceeded 80 °C. This was related to the acceleration of the rearrangement of double helices of starch molecules with the elevating temperature, and hence the gelatinization of starch was accelerated and the gelation strength was increased. Besides, the interaction of starch and protein also contributed to the changes of gelation strength at different temperatures. The heat-induced protein molecule was in a “melting sphere” state between the natural state and denaturation state [2,34], which intensely interacted with starch molecules at higher temperatures, and thus enhanced their gelation strength.

## 5. Conclusions

During the heating of the wheat flour suspension, high-molecular-weight protein polymers were formed. Covalent bonds (such as disulfide bonds) and non-covalent bonds (such as ionic bonds, hydrogen bonds and hydrophobic interactions) mainly participated in the formation of protein aggregates. Meanwhile, SDS-PAGE, intrinsic fluorescence and FTIR measurements showed that wheat gluten protein unfolded during thermal treatment and aggregated into a three-dimensional network when the temperature exceeded 70 °C. Considerable disruption of ordered structures of starch occurred with heating at 70 °C, and starch was fully gelatinized at 80 °C. Though the starch short-range ordered structure existed at higher temperatures, the IR ratio of 1047/1022 cm^−1^ decreased at higher temperatures. Heat treatment decreased the pasting viscosities of the cooked flour from control to 80 °C, while the viscosity of the cooked wheat flour increased with heating treatment at 90, 95 and 100 °C. Moreover, the formation of wheat flour gelation was ascribed to the intense interaction of starch and protein at higher temperatures, and the textural test showed that gelation formed by cooked flour suspensions at 90, 95 and 100 °C had higher strength. The information obtained from this study may provide a basis and a more useful guide for improving wheat flour processing conditions and the processing properties of wheat flour-based food products than might be obtained from studies of isolated protein or starch.

## Figures and Tables

**Figure 1 foods-10-01419-f001:**
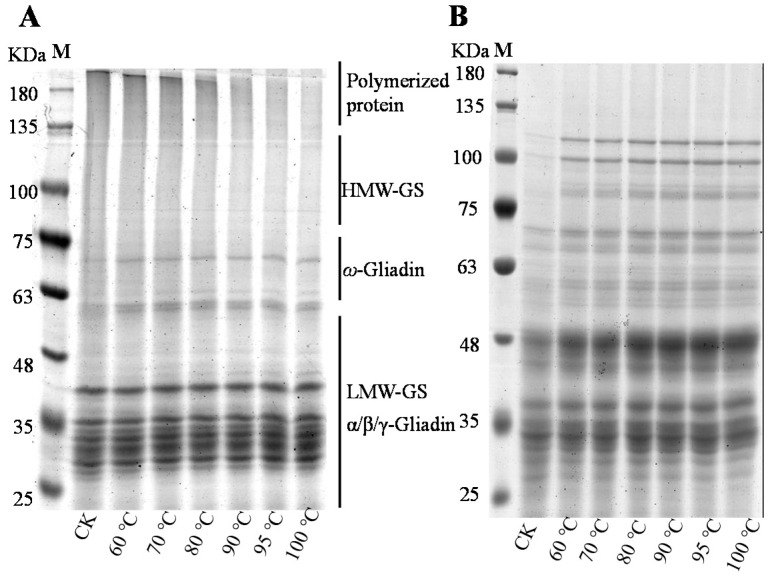
Non-reduced (**A**) and reduced (**B**) SDS-PAGE profiles of protein with different temperature treatment. CK, control; HMW-GS, high-molecular-weight glutenin subunit; LMW-GS, low-molecular-weight glutenin subunit.

**Figure 2 foods-10-01419-f002:**
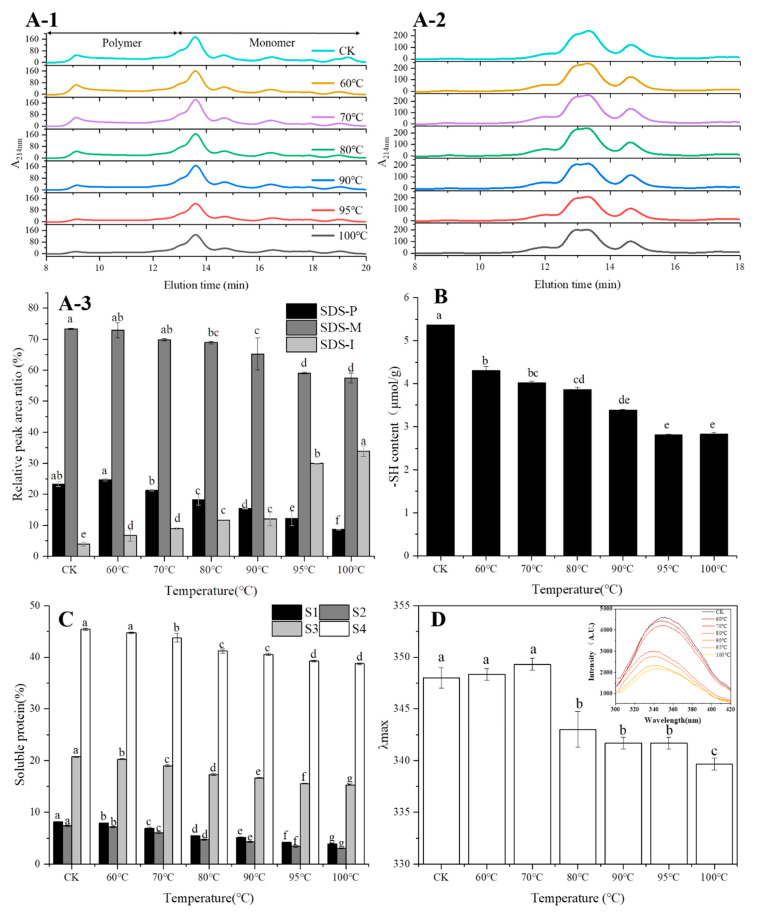
Typical non-reduced (**A-1**) and reduced (**A-2**) SEC-HPLC profiles, and relative amounts of SDS-soluble polymers, monomers and insoluble proteins (**A-3**) with different temperature treatment. Changes of free SH (**B**) and noncovalent bonds (**C**) content with different temperature treatment. Changes in intrinsic fluorescence spectra and λmax (**D**) of protein with different temperature treatment. The different letters indicate significant difference at 0.05 level.

**Figure 3 foods-10-01419-f003:**
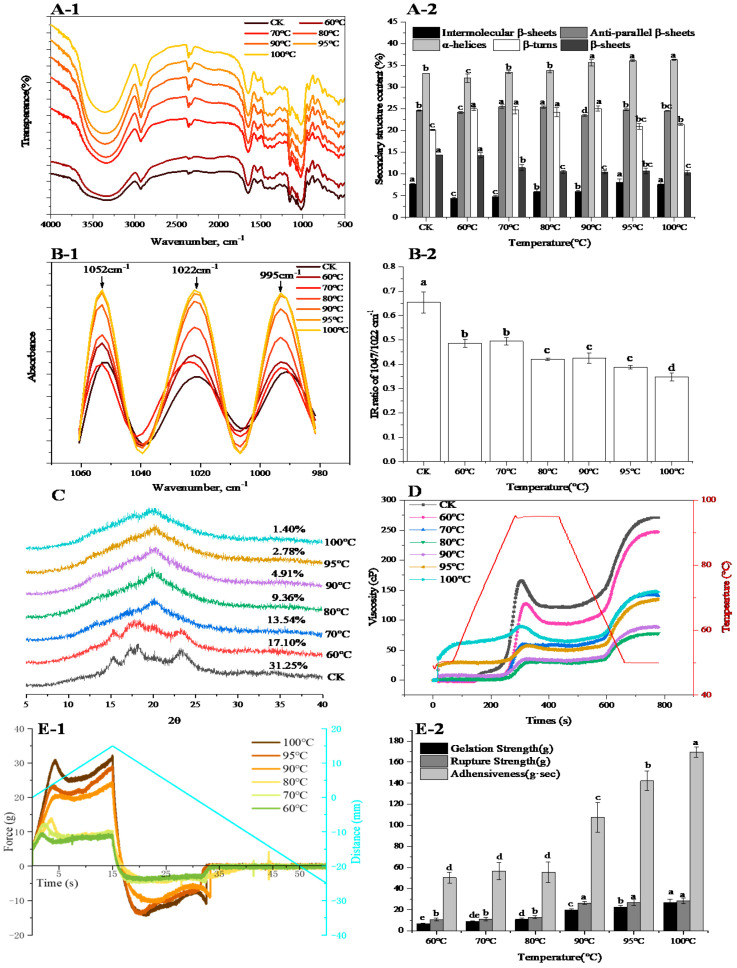
The deconvoluted original FTIR spectra in the region from 4000 cm^−1^ to 500 cm^−1^ (**A-1**) and secondary structure of protein (**A-2**) during heating of wheat flour. The deconvoluted FTIR spectra in the region from 1060 to 980 cm^−1^ (**B-1**) and the IR ratios of bands at 1047 and 1022 cm^−1^ of FTIR spectra (**B-2**). X-ray diffractograms of starch at different temperatures (**C**). RVA profiles at different heating temperatures of flour (**D**). Texture profiles (**E-1**) and texture parameters (**E-2**) of gelation formed with different temperature treatments of flour suspension. The different letters indicate significant difference at 0.05 level.

**Figure 4 foods-10-01419-f004:**
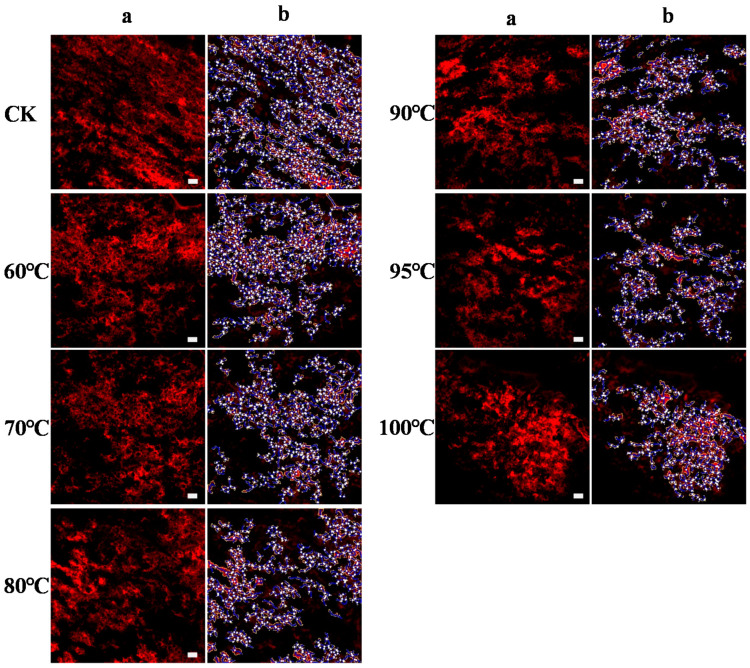
Morphological properties of protein network during heating of wheat flour obtained by CLSM. (**a**) images of protein network captured by CLSM, (**b**) images processed by Angiotool to visualize the protein network. Size bar = 20 μm.

**Table 1 foods-10-01419-t001:** Image analysis of CLSM of wheat protein in dough of different heating treatments.

	Control	60 °C	70 °C	80 °C	90 °C	95 °C	100 °C
Junction density (×10^−3^)	1.33 ± 0.03 c	1.62 ± 0.46 c	1.73 ± 0.43 bc	1.91 ± 0.18 bc	1.78 ± 0.01 bc	2.34 ± 0.46 ab	2.67 ± 0.18 a
Total protein length (μm)	6.68 ± 0.91 d	7.73 ± 0.49 d	9.39 ± 0.87 c	10.19 ± 0.64 c	10.56 ± 0.43 bc	11.72 ± 1.33 b	15.71 ± 1.04 a
Branching rate (×10^−2^)	0.75 ± 0.01 e	0.76 ± 0.01 e	0.78 ± 0.01 de	0.81 ± 0.01 cd	0.83 ± 0.02 c	0.87 ± 0.03 b	0.95 ± 0.05 a
Lacunarity (×10^−2^)	11.64 ± 0.67 c	17.85 ± 2.32 bc	21.88 ± 2.57 b	26.09 ± 3.65 b	26.64 ± 1.56 b	41.63 ± 11.45 a	43.61 ± 8.59 a
Protein area (μm^2^)	84.79 ± 2.71 a	68.08 ± 8.11 b	56.31 ± 1.21 c	54.31 ± 1.01 cd	49.42 ± 0.69 d	37.78 ± 5.17 e	39.51 ± 0.63 e

Different letters in the same row indicate a significant difference (*p* < 0.05).

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
