# Peer review of "Investigation on the Molecular and Physicochemical Changes of Protein and Starch of Wheat Flour during Heating"

_foods, 2021, doi:10.3390/foods10061419_

Round 1
Reviewer 1 Report
Very interesting paper - see my comments/suggestions in the attached file

Author Response
Dear Editor,
Thank you very much for giving us an opportunity to revise our manuscript, we also appreciate you and the reviewers’ valuable comments on our manuscript. Those comments are very helpful for revising and improving our paper. We have revised the manuscript carefully to address reviewers’ comments. Revised parts are marked in red in the paper. The main corrections in the paper and the responses to the reviewer’s comments are as following:
Point 1: For how long time? Lack information about time of heat treatment
Response 1: Wheat flour suspensions were incubated at 60, 70, 80, 90, 95 and 100 °C for 30 min. Now we added the time information of heat treatment in line 100.
Point 2: Unclear sentence. What do you mean by ingredients? Is is e.g. fat, vitamins, minerals etc in the flour OR added ingredients in the food matrix?
Response 2: Yes, the ingredients in present study mean fat, vitamins, minerals etc in the native flour, not “added ingredients”. Now we revised it and made it clear in line 354-355. Thank you for your advice, which makes our discussion more complete.
Point 3: Comment to Editor: It is only reference 1-8 that is mentioned in the introduction. Reference 9-15 are mentioned in the material/method section. Reference 16-34 are introduced for the first time in the discussion. I know different jounals have different policy concerning that. Is it ok with so many papers introduced for the first time in discussion or should they be introduced in the introduction instead?
Response 3: Thanks for your advice, we revised introduction and discussion carefully. References [2], [5], [6], [12] were cited both in introduction and discussion.
Other advices were also revised and marked in red colour.
We tried our best to improve the manuscript and hope that the correction will meet with approval.
Once again, thank you very much for your comments and suggestions.
Yours sincerely,
Qin Zhou
Reviewer 2 Report
In this work the authors perform a study to investigate the behavior of starch and protein of wheat flour during heating by means of several analytical experiments. The aim of the authors is the investigation of the physicochemical and molecular changes of these systems to improve wheat flour processing.
In my opinion, minor revision is needed (corrections to minor methodological errors and text editing). In fact, this study concerns a topic whose research began several years ago but is still in progress. Even if the work is written adequately and well organized in all its parts, I propose to the authors to add some more information that can help in the reading of the paper (e.g., details about the gliadin and the difference with glutenin; the importance to use the Shodes Protein KW-804 column in the SE-HPLC experiment, etc.). In particular, I suggest to improve the introduction and the method sections adding some details, for example, on the techniques, focusing on the importance on using these kind of experiments to reach the aim of the paper and to increase the reader's interest and curiosity. A further reading of the text to avoid typos and to make an appropriate use of English is also required.
Finally, I propose to the authors to deepen the study about the denaturation process of proteins through the use of FTIR as shown, for example, in the paper "The protein irreversible denaturation studied by means of
the bending vibrational mode", Physica A 412 (2014).
Author Response
Dear Editor,
Thank you very much for giving us an opportunity to revise our manuscript, we also appreciate you and the reviewers’ valuable comments on our manuscript. Those comments are very helpful for revising and improving our paper. We have revised the manuscript carefully to address reviewers’ comments. Revised parts are marked in red in the paper. The main corrections in the paper and the responses to the reviewer’s comments are as following:
Point 1: Even if the work is written adequately and well organized in all its parts, I propose to the authors to add some more information that can help in the reading of the paper (e.g., details about the gliadin and the difference with glutenin; the importance to use the Shodes Protein KW-804 column in the SE-HPLC experiment, etc.).
Response 1: Thanks for your advice, firstly, the difference between gliadin and glutenin was added in introduction (line 31-40). Secondly, the Shodes Protein KW-804 column used in the SE-HPLC experiment, is suitable for the analysis of macromolecular substances with molecular weights ranging from thousands to millions, such as proteins. The related information has been added in 2.3.1 section (line 110-111).
Point 2: In particular, I suggest to improve the introduction and the method sections adding some details, for example, on the techniques, focusing on the importance on using these kind of experiments to reach the aim of the paper and to increase the reader's interest and curiosity
Response 2: Thanks for your advice. Many advanced technologies and methods (FTIR, SE-HPLC, XRD, CLSM, etc.) have been used in this study to probe the dynamic changes of protein and starch during heating. The details about these technologies have been added in introduction (line 49-54) and the method section (line 161-164, 172-174) to address the importance of these technologies.
Point 3: I propose to the authors to deepen the study about the denaturation process of proteins through the use of FTIR as shown, for example, in the paper "The protein irreversible denaturation studied by means of the bending vibrational mode", Physica A 412 (2014).
Response 3: Thanks for your advice. O–H stretching of inter hydrogen-bond (3400 cm-1), C–H stretching of methylene (2950 cm-1), CH and CH2 stretching in cellulose (2850 cm-1), ester carbonyl group (1746 cm-1) were analyzed from FTIR spectrum to probe the functional groups involved in the denaturation process of proteins during heat processing. The details of FTIR spectrum have been described in 3.5 section (line 275-287) to deepen the study about the dynamic changes of chemical bonds or functional groups during heating.
Other changes in the manuscript were also marked in red color.
We tried our best to improve the manuscript and hope that the correction will meet with approval.
Once again, thank you very much for your comments and suggestions.
Yours sincerely,
Qin Zhou